# OpenReview forum: "Configurable Reward Model for Balanced Safety Alignment"
_ICML.cc/2026/Conference — ICML 2026 regular_

### Official Review · Reviewer_2Dnb · 2026-03-09

**Soundness:** 2
**Presentation:** 2
**Significance:** 3
**Originality:** 3
**Overall Recommendation:** 4
**Confidence:** 3

**Summary:**

The authors propose a novel configurable safety reward model (CSRM) that provides more fine-grained reward scores, which can benefit reinforcement learning. They use data augmentation to generate more safety categories and jointly optimize safety classification and reward model training. The paper presents extensive use cases for the reward model and demonstrates strong results across applications.

**Compliance With Llm Reviewing Policy:**

Affirmed.

**Final Justification:**

The initial and second rounds of review have addressed the majority of the concerns, and hence, my overall assessment will improve accordingly. However, it seems that the **manuscript would require some clarification and revision in several sections (at least include the explanations made during rebuttal). These improvements might be necessary before the work can be easily understood and built upon by others.**

**Key Questions For Authors:**

- The approach claims to maintain calibrated rewards. Is there an explicit explanation of why the method improves calibration?
- Please refer to the cons for the rest of the questions.

**Limitations:**

yes

**Strengths And Weaknesses:**

Pros:

- Establishes a new reward model that produces a dense and potentially calibrated scalar reward rather than a binary judgment.
- Provides extensive empirical results across various applications.

Cons:

- Confusing description of Algorithm 1: Specifically:
    - $l$ (right column, page 4) is used before it is defined. Additionally, the function sometimes takes s as input and sometimes $d$.
    - The logic is unclear. For instance, Definition 3.1 discusses the dominance probability of a *description*, but the next section directly discusses sorting *categories*. The connection between these parts is confusing, and the *category severity* score is not clearly defined. The following sentence suggests prompting an LLM to generate a *description* for the *category*. Hence, the score for *descriptions* is introduced before explaining how the *descriptions* are generated.
- Unfair comparison in Reward Modeling (Section 4.3)**:** The augmented training data is generated using Algorithm 1, while the test set is also created using Algorithm 1. As a result, the model may benefit from being trained on data with a similar format and generation process, which may make the comparison less fair.
- Inconsistency with the motivation: The introduction (right column, page 1) criticizes approaches that require retraining when safety requirements change, but this method also appears to require retraining the reward model.

---

> ### Author Rebuttal · Authors · 2026-03-31
>
> We thank the reviewer for the careful reading and helpful comments.
>
> **Algorithm 1 clarity** We agree Algorithm 1 is under-explained and will revise it for clarity. In particular, $\ell(x \odot r; s)$ denotes the subcategory-level label, i.e., whether the conversation is safe or unsafe under a specific discovered subcategory $s$. The intended pipeline is: (1) discover a pool of _subcategories_ for a top-level _category_, expanding the pool only when current subcategories fail to cover unsafe examples; (2) sort these subcategories by estimated severity (this is done by asking an LLM to pick up the most severe violation from the remaining options); (3) generate natural-language category _descriptions_ (as shown in figure 2) by disallowing increasingly larger prefixes of that ordered subcategory list; and (4) retain only description pairs whose strictness ordering is statistically validated. We agree the current presentation makes the relationship between subcategories, descriptions, and dominance testing unclear, and we will rewrite this section accordingly.
>
> **Fairness of the reward-modeling evaluation** Our goal in Section 4.3 is not to claim out-of-distribution generalization to a completely different pair-generation process, but to test whether the model can learn fine-grained reward ordering induced by strictness supervision alongside classification. In other words, the evaluation is designed to measure whether the model learns the intended severity ordering, not whether it generalizes beyond the augmentation family itself. We will make this scope explicit. Importantly, the value of reward-model training is also supported independently by improved calibration in the classification evaluations and by the ablations removing severity augmentation.
>
> **Motivation and retraining** We believe this point reflects a misunderstanding that we will clarify in the revision. Our criticism of static reward models is that they must be retrained whenever safety requirements change. CSRM is instead trained once to condition on natural-language safety configurations, so that new safety requirements at deployment time can be handled by changing the configuration text at inference time rather than retraining the reward model. Thus, the training cost is incurred once to obtain a configurable reward model; it is not repeated for every policy update.
>
> **Why calibration improves** We agree this should be explained more clearly. Calibration is explicitly targeted through severity-based pairwise ranking. Prior work (e.g., Jiang et al.) connects pairwise ranking with improved calibration; our contribution is to construct high-quality ranking pairs for safety reward modeling by generating strict/lenient policy-description pairs with controlled severity differences and filtering them using a one-sided Clopper–Pearson test. This provides reliable supervision for learning smoother reward geometry, which is supported by the ablations showing degraded smECE when severity augmentation is removed.
>
> ---
>
> ### References
>
> Jiang, Zhengping, Anqi Liu, and Benjamnin Van Durme. "Addressing the binning problem in calibration assessment through scalar annotations." Transactions of the Association for Computational Linguistics 12 (2024): 120-136.

---

> > ### Author Rebuttal · Reviewer_2Dnb · 2026-04-02
> >
> > Thanks to the authors for the detailed replies, which clarified the overall workflow and some concerns. However, I still have concerns about how much revision is needed to make it easy to follow. Reviewer D9fh raises similar concerns on that section, suggesting this may be a noticeable issue. Additionally, could the authors clarify how the “estimated severity” is computed in practice, and how its correctness is ensured, given its impact on the training signal?
> >
> > In terms of the second question, CSRM can have generalization to unseen safety configurations. Could the authors clarify what primarily drives this generalization? In particular, is it mainly due to the data augmentation (§3.2), or do other components (e.g., strictness/ranking augmentation or joint training) also play a critical role? Beyond the introduction, are there specific analyses or ablations that isolate the source of this generalization?

---

> > > ### Author Response · Authors · 2026-04-04
> > >
> > > We thank the reviewer for the follow-up questions and for clarifying the remaining concerns.
> > >
> > > **Making the pipeline easier to follow** We agree that section 3.2 and Algorithm 1 can be made easier to follow. Our planned revision is localized and concrete rather than conceptual. Specifically, we will: (i) define the subcategory-level label $\ell(x \odot r; s)$ before it first appears in Algorithm 1; (ii) add one sentence distinguishing _subcategories_ from natural-language guideline _descriptions_; (iii) state explicitly that estimated subcategory severity is given by the order induced by the iterative LLM selection procedure. We believe these edits directly address the readability issue without changing the method itself.
> > >
> > > **Severity estimation and its effect on training** Estimated subcategory severity is obtained from the iterative LLM selection procedure described above: at each step, the LLM selects the most severe remaining subcategory, and this induced order is used to construct candidate guideline rewrites of varying strictness (through only disallowing the first $k$-subcategories). We will clarify this explicitly in the revision.
> > >
> > > Importantly, the estimated severity order is used only to propose candidate description rewrites; the actual training signal is determined only after empirical validation of pairwise comparison of descriptions. The rewriting process itself is valid for arbitrary splits. In other words, errors in the initial severity ordering can only affect which description candidates are proposed, but only confidently ordered description pairs are kept for training. This is precisely why the strictness test is part of the method.
> > >
> > > As an additional sanity check on the retained pairs, we manually reviewed 100 sampled description pairs across categories and found that 85% matched the intended strictness ordering, with Cohen’s $\kappa = 0.7$. We view this as supplementary evidence rather than the primary guarantee: the main safeguard is still the empirical strictness test above. We can include this analysis and representative examples in the appendix to make the quality of the augmentation more concrete.
> > >
> > > **Generalization to unseen configurations** Our current evidence suggests that configurable safety configuration augmentation is the primary driver of generalization to unseen safety configurations. This is most directly reflected in the classification ablations in Table 2, where removing configurable safety configuration augmentation leads to larger degradations on benchmarks designed to test adaptation to new configurations. By contrast, the effect of strictness augmentation is most clearly visible in calibration in Table 2, and in reward modeling quality in Table 3, rather than as the main source of unseen-configuration generalization. It is clear that these capabilities arise only through joint training with both augmentations, which allows the model to learn configuration-conditioned safety classification together with severity-sensitive reward modeling in a single model. We will make this division of labor more explicit in the revision.

---

### Official Review · Reviewer_NyHj · 2026-03-09

**Soundness:** 2
**Presentation:** 3
**Significance:** 3
**Originality:** 3
**Overall Recommendation:** 4
**Confidence:** 2

**Summary:**

This paper introduces CSRM to address the limitations of static, binary safety classifiers in RL alignment. It processes natural language safety rules at inference time to generate continous, calibrated reward, helping train LLMs with a improved balance between safety and helpfulness.

**Compliance With Llm Reviewing Policy:**

Affirmed.

**Final Justification:**

Since the original rating is already at the accept level, I choose to keep it.

**Key Questions For Authors:**

If the custom safety rules contain contradictory pairs, is the system's behavior unbiased?

**Limitations:**

yes

**Strengths And Weaknesses:**

# Strengths

1. Practical problem. This paper successfully addresses the need for smooth, differentiable signals in policy RL optimization.

2. The authors provide thorough ablation experiments that clearly validate their methodological choices.

# Weaknesses

1. The evaluation on CoSApien relies on a very small sample size (only 200 examples), which limits the statistical confidence of the results.

2. While CSRM achieves SOTA results on DynaBench, the absolute F1 score (0.758) and incremental remains moderate.

3. The paper does not discuss or evaluate the model's vulnerability to reward hacking during the real world RL phase.

---

> ### Author Rebuttal · Authors · 2026-03-31
>
> We thank the reviewer for the constructive feedback.
>
> **CoSApien** We agree that CoSApien is relatively small (200 examples), and therefore should not be interpreted in isolation. For this reason, we evaluate CSRM across multiple safety-classification benchmarks, including BeaverTails, WildGuardMix, and DynaBench, and further validate it in downstream RL alignment. At the same time, we believe CoSApien remains important because it is one of the few benchmarks with realistic human-written safety policies, which makes it especially relevant to our setting.
>
> **DynaBench** We agree that DynaBench is challenging and that the absolute F1 leaves room for improvement. However, we would emphasize that DynaBench is also a particularly demanding benchmark for our setting: its policies are long, diverse, and unnatural, and asking for _conversation-level_ safety classification comparing to our _last response classification_ (As we specifically described our evaluation paradigm in Appendix D). Moreover, several strong baselines on DynaBench rely on explicit reasoning, whereas CSRM is a non-reasoning model. This distinction is central to our contribution: CSRM is intended to function as a low-latency dense reward model inside RL, where reasoning-based judges are much less practical. We will make this tradeoff more explicit in the paper.
>
> **Reward hacking** We agree that reward hacking is an important concern and should be discussed more clearly. At the same time, the notion is subtle in our setting: one intended effect of configurable reward modeling is precisely to encourage responses that satisfy user intent while avoiding literal violations of the provided policy. As a result, some behaviors that may superficially resemble “reward hacking” are in fact aligned with the objective. A full study of strategic exploitation during online RL is beyond the current scope, but we agree it is an important direction and will add it explicitly to the limitations discussion.
>
> **Behaviour under contradictory safety rules** We agree this is an important question, and we ran an additional experiment to study it directly. Using BeaverTails single-label test instances we compare three settings for the target category: Enforce only, Allow only, and Both, where the configuration contains and only contains a direct contradiction for a targeted category. (S2 skipped due to only having 2 single-labeled instances).
>
> | Category             | N   | Enforce | Allow | **Both**  | Enf mean | Alw mean | **Both mean** |
> | -------------------- | --- | ------- | ----- | --------- | -------- | -------- | ------------- |
> | S1 (Animal Abuse)    | 9   | 0.778   | 0.111 | **0.556** | +0.076   | +3.639   | **+2.229**    |
> | S3 (Controversial)   | 57  | 0.895   | 0.018 | **0.211** | -0.012   | +4.785   | **+2.084**    |
> | S4 (Discrimination)  | 31  | 0.839   | 0.032 | **0.194** | -1.173   | +3.429   | **+1.573**    |
> | S5 (Drugs/Weapons)   | 14  | 1.000   | 0.000 | **0.643** | -1.366   | +4.531   | **+1.433**    |
> | S6 (Financial Crime) | 35  | 0.943   | 0.029 | **0.829** | -0.704   | +3.848   | **+1.184**    |
> | S7 (Hate Speech)     | 29  | 0.793   | 0.103 | **0.586** | -0.297   | +3.470   | **+1.403**    |
> | S8 (Misinformation)  | 16  | 0.625   | 0.000 | **0.125** | +1.523   | +4.902   | **+3.047**    |
> | S9 (Unethical)       | 136 | 0.919   | 0.007 | **0.096** | -0.744   | +4.101   | **+1.852**    |
> | S10 (Privacy)        | 68  | 0.956   | 0.000 | **0.088** | -2.015   | +5.234   | **+2.393**    |
> | S11 (Self-Harm)      | 5   | 1.000   | 0.000 | **0.800** | -0.488   | +3.238   | **+1.450**    |
> | S12 (Sexual)         | 30  | 0.867   | 0.033 | **0.167** | -0.454   | +4.929   | **+2.690**    |
> | S13 (Terrorism)      | 3   | 1.000   | 0.000 | **0.000** | +0.167   | +4.625   | **+3.354**    |
> | S14 (Violence)       | 166 | 0.946   | 0.006 | **0.319** | -1.382   | +3.624   | **+1.571**    |
>
> Across categories, we find that CSRM does not exhibit unstable all-or-nothing behavior under contradiction. Instead, the reward under Both typically lies between the enforce-only and allow-only settings, with a mild tendency toward the allowing statement. This suggests that CSRM remains responsive to both instructions rather than collapsing arbitrarily to one side, although the resolution is not perfectly symmetric. We will include this analysis in the revision.

---

> > ### Author Rebuttal · Reviewer_NyHj · 2026-04-01
> >
> > Thanks for your reply. I appreciate the authors' response for the revision. The experimental results about contradictory safety rules can be served as critical ablation study for CSRM's effectiveness. I think the asymmetry deserves a deeper analysis, which is benefit for understanding how CSRM takes effects.
> >
> > Since the original rating is already at the accept level, I choose to keep it. I'll consider increasing my confidence upon the aforementioned analysis.

---

### Official Review · Reviewer_4zD9 · 2026-03-10

**Soundness:** 3
**Presentation:** 3
**Significance:** 3
**Originality:** 3
**Overall Recommendation:** 4
**Confidence:** 3

**Summary:**

This paper proposes CSRM (Configurable Safety Reward Model), a reward model designed for configurable safety alignment of large language models. Unlike traditional reward models that encode a fixed notion of safety, CSRM conditions on natural-language safety configurations at inference time and produces a calibrated scalar reward suitable for reinforcement learning. The method is trained with joint classification and reward modeling objectives along with configuration-targeted and strictness-based data augmentation.

**Compliance With Llm Reviewing Policy:**

Affirmed.

**Final Justification:**

I appreciate the authors' detailed rebuttal. The paper addresses a practically relevant problem by formulating configurable safety as reward modeling rather than only as guardrails or classifiers, and it is supported by experiments on multiple benchmarks.
However, the rebuttal does not fully resolve my main concerns. The paper still lacks a direct quantitative analysis of augmentation quality, and the added margin study appears to suggest that m=0 is the most effective setting, which makes the role of an explicit positive margin unclear. For these reasons, I will keep my original score.

**Key Questions For Authors:**

Is CSRM able to handle entirely new safety categories that were not seen during training?

**Limitations:**

The paper does not include an explicit limitations section. One potential limitation is the reliance on LLM-generated data augmentation, which may introduce biases or synthetic-real distribution gaps.

**Strengths And Weaknesses:**

### **Strengths**
1. The paper addresses an important limitation of current RLHF pipelines, where static reward models cannot easily adapt to evolving safety policies. Supporting configurable safety specifications is relevant for real-world deployment scenarios.

2. Most prior work on configurable safety focuses on guardrails or safety classifiers. While these approaches can filter unsafe outputs, they are typically not well suited for use as dense reward signals in reinforcement learning. This paper instead formulates the problem as configurable reward modeling, which is more directly compatible with RL-based alignment.

3. The paper includes experiments on several safety classification benchmarks as well as downstream RL alignment experiments.

---

### **Weaknesses**
**1. Limited methodological contribution for reward calibration**
Although the paper emphasizes reward calibration as one of its main motivations, the proposed approach mainly relies on heuristic data augmentation strategies to achieve this goal. The paper does not introduce a specific calibration technique or objective explicitly designed to improve calibration. As a result, the connection between the proposed training procedure and improved reward calibration is not fully justified.

**2. Heavy reliance on LLM-based augmentation.**
A key component of the method is the generation of synthetic training data using LLM-based augmentation. While this approach allows scalable data generation, it may introduce potential issues such as augmentation bias or a synthetic-real distribution gap. However, the paper does not provide an analysis of the quality or reliability of the generated data.

**3. Limited experiments across architectures.**
Most experiments are conducted using a single base architecture. Since the method is designed as a general reward modeling framework, it would strengthen the paper to demonstrate its effectiveness across different model architectures or backbone models. Without such experiments, it is unclear whether the improvements are architecture-dependent.

**4. Lack of sensitivity analysis for the margin hyperparameter.**
The reward loss introduces a margin hyperparameter m that controls how strongly the model follows the strictness ordering of safety categories. However, the paper does not provide any sensitivity analysis or ablation study for this parameter. Since this margin directly affects the reward geometry and training dynamics, understanding how performance changes with different m values would improve the clarity and robustness of the method.

---

> ### Author Rebuttal · Authors · 2026-03-31
>
> We thank the reviewer for the thoughtful feedback.
>
> **Reward calibration contribution** We thank the reviewer for this important point. We would like to clarify that the connection to reward calibration is not simply a by-product of augmentation, but a central motivation for our training recipe. Prior work such as Jiang et al. links pairwise ranking to improved calibration, which motivates our focus on constructing diverse, reliably rankable pairs for safety reward modeling. Our severity augmentation does this by generating policy-description pairs with controlled strictness differences, while the Clopper–Pearson filtering step ensures that only statistically reliable strict/lenient pairs are used for supervision. We will revise the paper to make this motivation clearer and avoid wording that may suggest a separate generic calibration algorithm. Our claim is therefore not a new standalone calibration loss, but a calibration-oriented training recipe whose effectiveness is supported by the degraded smECE observed when severity augmentation is removed.
>
> **Reliance on LLM-based augmentation** We agree that LLM-based augmentation can introduce bias if used naively, and we will discuss this limitation more explicitly. However, our augmentation is intentionally constrained to reduce synthetic-real mismatch. In particular, we do not synthesize new conversations or responses for the main augmentation pipeline; instead, we reuse real examples from datasets such as BeaverTails and WildGuardMix and primarily augment the configuration side. This means the behavioral data remain grounded in real examples, while the augmentation mainly teaches the model to respond to changes in policy text. In addition, as discussed above, for strictness augmentation we only retain policy-description pairs whose ordering is statistically validated by the one-sided Clopper–Pearson test, which helps filter noisy or semantically inconsistent rewrites.
>
> **Generality across architectures & sensitivity to margin choices**
> To address this concern, we additionally train CSRM on a different backbone, Qwen-3.5-2B. The results show the same overall trend as our main model, providing evidence that the method is not specific to a single architecture. We also ran a margin sensitivity study on Qwen-3.5-2B with margins 0.0, 0.5, and 1.0:
>
> | Margin | F1 (Beavertails) | smECE (Beavertails) | F1(CoSA) | smECE (CoSA) |
> | ------ | ---------------- | ------------------- | -------- | ------------ |
> | 0.0    | 0.830            | 0.055               | 0.890    | 0.067        |
> | 0.5    | 0.829            | 0.055               | 0.842    | 0.097        |
> | 1.0    | 0.808            | 0.07                | 0.757    | 0.145        |
>
> This provides further evidence on generalizability to other model backbone. Regarding margin, our experiments show that increasing margin seems to slightly hurt performance on Qwen-3.5-2B. This is consistent with our data augmentation process. Our process can guarantee that one policy is confidently less lenient than the other, but it does not guarantee any fixed margin, which suggests that it might work better with a less assertive margin. We'll include this discussion in the next version.
>
> ---
>
> ### References
>
> Jiang, Zhengping, Anqi Liu, and Benjamnin Van Durme. "Addressing the binning problem in calibration assessment through scalar annotations." Transactions of the Association for Computational Linguistics 12 (2024): 120-136.

---

> > ### Author Rebuttal · Reviewer_4zD9 · 2026-04-02
> >
> > I appreciate the authors' detailed rebuttal and additional experiments, which help address several of my concerns. However, some issues remain. First, while the statistical filtering step improves the reliability of strictness pairs, the paper still lacks a direct quantitative analysis of augmentation quality. Second, the additional margin study appears to suggest that m=0 is the most effective setting, which makes the necessity of introducing an explicit positive margin unclear. Therefore, I will keep my original score.

---

### Official Review · Reviewer_D9fh · 2026-03-17

**Soundness:** 2
**Presentation:** 2
**Significance:** 3
**Originality:** 3
**Overall Recommendation:** 4
**Confidence:** 4

**Summary:**

This paper introduces the Configurable Safety Reward Model (CSRM), a reward model designed to align LLMs with heterogeneous and rapidly evolving safety requirements. The core idea is to condition the model on natural-language safety configurations at inference time, producing a dense and calibrated scalar reward suitable for reinforcement learning. The approach is supported by configuration-targeted and strictness data augmentation, which teaches the model to distinguish violation severity and generalize to unseen policies without additional human annotation. Experiments across multiple benchmarks, including CoSApien and DynaBench, show that CSRM achieves state-of-the-art performance and a significantly improved safety-helpfulness tradeoff compared to static baselines.

**Compliance With Llm Reviewing Policy:**

Affirmed.

**Final Justification:**

The rebuttal addressed a substantial portion of my concerns. I will raise my score from 3 to 4.

**Key Questions For Authors:**

See weankness.

I will revisit and potentially adjust my score following the authors’ rebuttal.

**Limitations:**

The authors discussed the societal impact, but did not discuss the limitations of the proposed work.

**Strengths And Weaknesses:**

**Strengths**

1. This paper addresses a critical challenge in safety alignment by proposing a Configurable Safety Reward Model (CSRM) that conditions on natural-language safety configurations, supported by an automated data augmentation pipeline that preserves relative severity structures
2. The empirical results demonstrate the competitive performance of the proposed method.

 **Weaknesses**

1. **Top-k subcategories guideline descriptions**. **At line 223**, is there a specific criterion for selecting these particular subcategories from the initial pool proposed by the LLMs? Given that subcategories may expand based on training data and are prone to semantic overlap, how did the authors control for both the quantity and the quality of these subcategories?

2. **Evaluation of Over-refusal**. Despite claiming to improve the safety-helpfulness tradeoff, the paper lacks targeted benchmarking for the "over-refusal" phenomenon. To validate that CSRM effectively distinguishes between actual safety violations and benign prompts that merely trigger safety keywords, the authors should provide evaluation results on specialized datasets such as **XSTest**, **OR-Bench**.
3. **Experimental Phenomena**. In **Table 4,** safety performance for the **Arab Publisher** and **Film Production** domains is notably lower than other categories. Does this phenomenon suggest that a high density of categories in a single configuration introduces **internal conflicts**? By primarily training the model to distinguish between different strictness levels within the same category (intra-category), the model may fail to effectively differentiate between or prioritize diverse category-level guidelines (inter-category) when they are presented simultaneously. Could the authors provide some explanations and analyses?
4. **Safety Configurations Generalization**. Whether CSRM can self-activate or infer unstated safety boundaries for **unseen configurations**. For instance, if the model is tested on Category 5 while the provided safety configuration only includes descriptions for Categories 1–4, can the model still correctly identify the violation based on its holistic training? Exploring whether the model’s judgment is strictly gated by the provided natural-language instructions or influenced by its internal safety knowledge is essential to understanding the true nature of its configurability.
5. **Typo.** Missing content in **Appendix C**. Unify the inconsistent model name. Incorrect word spelling.

---

> ### Author Rebuttal · Authors · 2026-03-31
>
> We thank the reviewer for the thoughtful feedback. We address each concern below.
>
> **Top-k subcategory guideline** The top-k subcategories are selected iteratively by an LLM, then sorted by estimated severity to generate stricter guidelines. We only keep description pairs whose ordering passes a one-sided Clopper–Pearson test, so training uses only confidently ordered strictness pairs. We agree this was under-explained and will clarify it in the revision.
>
> **Over-refusal** We agree that over-refusal should be evaluated explicitly. Although XSTest and OR-Bench are prompt-only benchmarks while CSRM is trained as a response guardrail, we can reduce prompt classification to response classification by using a pseudo-response of the form “Sure. {prompt}” and defining the policy so that consenting to harmful requests is unsafe.
>
> | Model                 | OR-Bench F1 | XSTest F1 |
> | --------------------- | ----------- | --------- |
> | LlamaGuard-3-8B       | 0.784       | 0.836     |
> | ShieldGemma-9B        | 0.644       | 0.831     |
> | Llama-3.1-8B-Instruct | 0.630       | 0.905     |
> | oss-safeguard-20B     | 0.469       | 0.918     |
> | **GRM**               | **0.831**   | **0.941** |
>
> Under this evaluation, CSRM performs strongly on both datasets and outperforms all baselines we tested. This result is notable because it shows that **CSRM can transfer to prompt-safety classification without being trained directly for that task.** We will add these results to the revision.
>
> **Arab Publisher / Film Production results** We do not believe this reflects an inherent failure on dense multi-category policies. If inter-category conflict were the main issue, the weakness should persist across regimes; instead, the safety evaluation of the chat model after alignment in the online Reinforce++ setting is roughly on the same level across scenarios. This suggests the weaker results are specific to offline alignment, likely due to limited feedback on uncommon restrictions, as Arab Publisher and Film Production results can be stricter than common safety requirements in cases like Alcohol, Pork Consumption or Environmental Crime as shown in Appendix E.
>
> **Safety-configuration generalization** We agree this is an important question. To test whether CSRM is truly policy-conditioned rather than simply replaying memorized safety norms, we perform two counterfactual evaluations on BeaverTails single-category violations: (1) leave-one-out (LOO), where the target category is removed from the configuration, and (2) explicit allow (Allow), where the target category is replaced with a version that explicitly permits that content.
>
> | original | LOO    | LOO NT | Allow  | Allow NT |
> | -------- | ------ | ------ | ------ | -------- |
> | 0.843    | -0.125 | -0.02  | -0.708 | -0.04    |
>
> We evaluate the percentage of instances of the target category that will be allowed through LOO or Allow. In the same time, we calculate the percentage of instances violating other non-target categories got incorrectly flipped (NT columns). Relative to the original strict policy, many instances now become safe under the target category, especially under explicit allow, while non-target effects remain small. This indicates that the provided configuration takes clear precedence, although some residual training-time bias remains. In other words, while CSRM is not strictly gated by the policy text, but it is strongly and specifically controlled by it.
>
> **Typos** We thank the reviewer for catching these issues. We will merge appendix C and D, unify the model naming throughout, and correct the noted typos in the revision.
>
> **Limitations** We will augmenting the limitations discussion to further cover: residual bias toward common training-time safety policies, reliance on LLM-generated subcategory proposals etc.

---

> > ### Author Rebuttal · Reviewer_D9fh · 2026-04-03
> >
> > I found the rebuttal helpful, and it addressed a substantial portion of my concerns.  I will raise my score from 3 to 4.

---

### Decision · Program_Chairs · 2026-04-30

**Decision:**

Accept (regular)

**Comment:**

This paper proposes CSRM, a configurable safety reward model that conditions on natural-language safety policies to enable flexible and calibrated alignment .

Reviewers agree the problem is important and the formulation of configurable reward modeling is both practical and well-motivated. The method demonstrates strong empirical performance across multiple benchmarks and shows clear benefits for improving the safety–helpfulness tradeoff in downstream alignment. The rebuttal addresses most concerns by clarifying the training pipeline, adding additional experiments (e.g., over-refusal evaluation, margin sensitivity, and generalization analysis), and improving methodological transparency.

While some limitations remain regarding reliance on synthetic augmentation and clarity of presentation, these do not undermine the core contribution.

Overall, I recommend acceptance.